# Examining Care Assessment Scores of Community-Dwelling Adults in Flanders, Belgium: The Role of Socio-Psychological and Assessor-Related Factors

**DOI:** 10.3390/ijerph182211845

**Published:** 2021-11-11

**Authors:** Shauni Van Doren, David De Coninck, Kirsten Hermans, Anja Declercq

**Affiliations:** 1LUCAS Center for Care Research and Consultancy, KU Leuven, 3000 Leuven, Belgium; kirsten.hermans@kuleuven.be (K.H.); anja.declercq@kuleuven.be (A.D.); 2Center for Sociological Research, KU Leuven, 3000 Leuven, Belgium; david.deconinck@kuleuven.be

**Keywords:** community-dwelling adults, home care, BelRAI, contextual factors, assessor bias, Belgium, healthcare services

## Abstract

One of the primary objectives of health systems is to provide a fair system by providing a comprehensive and holistic approach to caregiving rather than focusing on a single aspect of a person’s care needs. This approach is often embodied by using standardized care assessments across health and social care settings. These assessments are completed by professional assessors and yield vital information regarding a person’s health or contextual characteristics (e.g., civic engagement, psychosocial wellbeing, environmental characteristics, informal care). However, these scores may be subject to bias that endangers the fairness of the health system. In this study, we investigate to what extent socio-economic and psychological indicators and assessor-related indicators are associated with BelRAI Screener care assessment scores amongst 743 community-dwelling adults nested within 92 assessors in Flanders, Belgium. Findings indicate that there is significant variance in scores at the assessor-level. Socio-psychological characteristics of clients are associated with scores: being fluent in Dutch and providing informal care are associated with low care dependency, while living with children, feelings of depression, and the presence of an informal caregiver during assessment are associated with high care dependency. We discuss the importance of rigorous assessor training and the potential for socio-psychological factors to contribute to the allocation of welfare benefits in light of the Flemish home care system’s potential (lack of) fairness.

## 1. Introduction

A steady increase in the prevalence of complex and often chronic care needs of people living at home in Flanders (the Dutch-speaking region of Belgium) confronts care providers and policymakers with important challenges [1,2,3,4,5]. Persons with complex care needs are often characterized by

a combination of comorbidities,mental health challenges and/orsocial vulnerability [6,7].

Gaining insight into the relationship between these three aspects across time, institutions, and regions supports the different stakeholders that are embedded in specific health systems in making informed decisions about how to address the complexities of maintaining quality and continuity of care [8].

Health systems have three independent objectives.

The primary objective is to improve health or prevent further decline (*health*).Second, they aim to provide a fair system of financial contribution (*financial fairness*).The third and final goal of health systems is to be responsive towards people’s expectations regarding dignity, autonomy and confidentiality of information (*responsiveness*) [9,10].

These three objectives collectively contribute to the fairness of a health system. Fairness means that the system responds equally well to everyone, without discrimination or differences in how people are treated [9].

### 1.1. Standardized and Comprehensive Care Assessments

One way in which a health system can meet the goal of fairness is by using standardized care assessments across health and social care settings. Care assessment scores can inform and improve care providers’ practice. For instance, re-assessment scores during and after an intervention can provide care providers with valuable information on the effectiveness of treatments and can facilitate comparisons with other clients who have complex and often chronic care needs [11]. Analysing the quality of data is important for all users of assessments and their output [12,13]. Examining to what extent care assessment scores vary between assessors is crucial to get an idea of potential assessor-related bias in the data. Furthermore, it is important to get a clear idea of which socio-psychological factors are associated with assessment scores to provide users of the data with reliable information about the socio-economic and socio-psychological context in which the person is living. This information may also serve to better train assessors and develop effective and fair care policies.

### 1.2. Threats to a ‘Fair Health System’

#### 1.2.1. Fragmentation in the Assessment of Care Needs

Even though a comprehensive and holistic approach to caregiving is recommended, many care providers, interventions and policies focus on one aspect of a person’s complex care needs [14,15,16,17,18], thus resulting in a fragmented caregiving approach. This approach is also accompanied by a variety of disease- or population-specific instruments to identify care needs. The drawbacks to primarily using disease-specific measures are that:the outcomes across populations are practically impossible to compare given the many diseases and the fact that a large proportion of people have multimorbidity and would need several, partly overlapping disease-specific instruments [19], andunderlying issues that go beyond the immediate, presumed problem are hard to detect [20].

Many studies emphasize the importance of the holistic approach that considers “the dynamic interplay between people and the surrounding social structures of a changing society” [20,21,22,23] (20, p. 689). Overlooking the interplay between people and their socio-economic context in research and care practices may result in inaccurate or unnecessary (health) care interventions. Research shows that contextual factors (e.g., socio-economic conditions, informal care context, et cetera) can elicit a moderating, mediating, independent or confounding effect on a persons’ (experiences of their) health (status) [22,24,25,26,27,28,29,30].

#### 1.2.2. Socio-Economic Determinants at the Client Level and the Care Context

Socio-economic determinants of health at the client level such as housing conditions, housing tenure and financial situation are unevenly distributed across populations and play a large role in the creation and maintenance of health inequalities [31,32]. These determinants affect both communicable and non-communicable diseases, making this a priority for anyone involved in health policy. The World Health Organization (WHO) has also highlighted the growing importance of socio-economic health determinants [31,32,33]. In 2005, they established a commission on the social determinants of health that reviewed the existing evidence, but also raised the societal debate and recommended policies to reduce these inequalities [33]. Research has primarily provided evidence on the association between (access to) health and objective economic inequality. In conjunction with the objective financial situation, there is, however, also a positive correlation between subjective financial well-being and (mental) health outcomes [34]. Also, a language barrier or a significant degree of illiteracy affects a person’s ability to access and use the necessary health care services [35,36].

The literature on the impact of providing care for family members on the health of the family caregivers is varied. We see both negative and positive effects of informal caregiving on the caregivers’ health. Caring for a person with chronic and/or complex care needs may result in caregiver distress, which in turn can negatively affect their health [37,38]. Nevertheless, research shows that this negative impact can be moderated by positive aspects of caregiving, such as experiencing the act of caregiving as rewarding [39]. In contrast to providing care, research also affirms the positive impact of receiving care from family members on a variety of patients’ physical and mental health outcomes [40,41].

#### 1.2.3. Assessor-Related Bias

Taking a holistic approach is indispensable in maintaining fairness in health systems, but the care instruments that are used are often completed by assessors (e.g., social workers). Although they are trained in or have experience with such instruments, this methodology also potentially endangers fairness, as assessors may complete assessments in different ways depending on individual or in addition to contextual characteristics, regardless of training. Thus, while (standardized) instruments can be developed with the goal of providing a holistic view of a person’s health status, the method of use by the assessors may endanger this objective.

Assessors frequently use a combination of data collection methods to complete a standardized (care) assessment. Examples of these methods are direct observations, surveys and semi-structured interviews. Hyman et al. [42] (p. 20) stated that “in social research, the measuring instrument is the interviewer”. As mentioned, quantifying care needs through a standardized assessment process has many advantages. Nevertheless, it is important to remember that all data collection methods introduce a certain amount of bias in the data. For example, the observer’s own beliefs will influence what is observed and reported. Furthermore, the presence of an observer or third party (e.g., family member) during an assessment can result in changes in the participant’s and assessor’s behavior [43]. Boyd Jr and Westfall [44] referred to the assessor as “a source of survey measurement error”.

The (perceived) quality of the data is important for all users of the assessment and its output. When validity and reliability studies do not properly incorporate potential assessor-related bias in their data, the users can question the data quality. This uncertainty can, in turn, create a laissez-faire attitude and an incorrect use of the assessment and output in practice [12,13]. Clustered datasets can help us identify interviewer variation or within-interviewer correlation. This occurs when the results of different respondents who are all assessed by the same interviewer are more alike than those of respondents who are assessed by different interviewers [45].

### 1.3. Research Questions

In this paper we want to investigate to what extent home care assessment scores are associated with socio-psychological and assessor-related factors. We use data from care assessment scores given by social care services in Flanders (Belgium) who are using the BelRAI Screener. We use data from home care assessments because the potential for bias in these data is high, given that home care clients often receive care and/or are assessed by various professionals from different home care organizations. This is unlike, for example, individuals living in residential care facilities. The exchange of information between these home care professionals is often lacking, meaning that this is potentially a feeding ground of assessor- or context-related bias. Social care at home (assistance with groceries, cleaning, finances, organizing the household and the care for children, et cetera) is a highly accessible service in Flanders, and acts as a gateway to more specialized care at home and aids in the allocation of care budgets. By providing hands-on assistance in these practical and re-occurring situations, professionals get a deeper understanding of the role of the environment on the functional status of a person.

This research is relevant in the Flemish context because home care in Flanders, Belgium is highly accessible, and a large proportion of clients have less complex problems. The current assessment methods in social care services were not sufficiently standardized or scientifically validated. Stakeholders such as social care services, patient organizations and policy makers collaborated to develop a validated short-form instrument to assess the biomedical aspects of functioning and the problems with activities of daily living, namely the BelRAI Screener [46,47]. The link between care needs and the socio-psychological context is becoming increasingly important in the academic literature on this topic and according to social care service organizations [2,14]. However, policy makers have found it difficult to translate these insights into a practically applicable instrument in the field. To address this issue, the relevant stakeholders have closely worked together to develop a new supplementary assessment that provides a clear overview of a person’s socio-psychological context: the BelRAI Social Supplement [48,49]. For these reasons, Flanders presents an ideal region to conduct this study given the recent development of these new instruments along with the growing attention of both policy makers and experts on the interplay of biological, psychological and social factors on a person’s care needs. To measure the extent of fairness of the assessments used in social care, we will use previously collected data to identify potential confounding factors in the scores.

The research questions are:To what extent do care assessment scores of community-dwelling adults in Flanders vary between assessors?How are socio-psychological factors of clients associated with care assessment scores among community-dwelling adults in Flanders?To what extent are assessor or assessment characteristics associated with care assessment scores among community-dwelling adults in Flanders?

## 2. Materials and Methods

### 2.1. Data Collection

To investigate the association of socio-psychological and assessor-related factors with assessment scores, BelRAI Screener (BRS) and BelRAI Social Supplement (BSS) data were collected among community-dwelling clients of social care services in Flanders, Belgium. Cross-sectional data were collected in 2019 by professional social workers whose job it is to (re-)assess care needs of individuals to check their eligibility for care benefits and, if needed, contact the appropriate care providers.

### 2.2. Sample

We included persons with chronic (physical and/or mental) diseases or disabilities, and excluded persons receiving maternity care and/or services for families in precarious situations. The reason for this is that these groups often require very specific care and support within a complex social context. We are aware that the current BelRAI Screener and the BelRAI Social Supplement cannot capture these very specific care needs. A person with care needs had to be of legal age (+18 years old), living at home, and be able to give their consent to be assessed for this study. Informed consent was obtained from all participants.

### 2.3. Instruments

#### 2.3.1. BelRAI Screener

The BelRAI Screener was developed in collaboration with Flemish stakeholders to evaluate care needs of adults in home care using five main questions and elaboration modules, namely:Instrumental Activities of Daily Living (IADL),Activities of Daily Living (ADL),Cognitive problems,Psychological problems, andBehavioral problems.

From June 2021 onwards, this BelRAI Screener has been used by all social care services in Flanders to get an efficient first assessment of a person’s care needs. It allows for the calculation of a dependency and care complexity index to determine whether a comprehensive care assessment is necessary. The dependency score also checks a person’s eligibility for a regional care budget [46]. All items of the BelRAI Screener originate exclusively from the existing and validated interRAI suite of assessment instruments [47].

#### 2.3.2. BelRAI Social Supplement

The BelRAI Social Supplement is a supplement to the BelRAI Screener and other interRAI instruments that gathers information on the social context of community-dwelling adults with care needs. The version of the BelRAI Social Supplement that was used in this study consisted of 101 items that were categorized into four themes:Environmental assessmentCivic engagementPsychosocial wellbeing; andInformal care and support [48,49]

### 2.4. Procedure

One hundred care professionals of organizations providing social care services were recruited to participate in the study. Six standardized training cycles on the two BelRAI instruments were organized across Flanders. Each training cycle consisted of a full day of training and three discussion groups. These discussion groups were organized at regular intervals after the day of training (approximately one month, three months and five months). They combined a teaching and feedback moment and participants were expected to attend at least one of these.

Following a maximum of four training sessions by the researchers, each assessor was asked to assess 10 clients with a BelRAI Screener [46] and the newly developed BelRAI Social Supplement [48] with the aim of variability in client profiles within the criteria mentioned above. These data were collected as part of a larger evidence-based policy research project to develop a Social Supplement to existing interRAI instruments “to assess the social context of community-dwelling adults with care needs” [50].

We programmed the BelRAI Screener and BelRAI Social Supplement instruments in Qualtrics, a cloud-based survey platform. An anonymous URL-link to the assessments was made available through a password-protected website with accompanying training materials. Professional caregivers also received a hard copy of both assessments when the appropriate hardware was not available to them. The assessors were asked to enter the information via computer, laptop, smartphone or tablet using a unique identifier. This identifier allowed us to link the client data to an assessor during data cleaning. Both assessments were completed during a single home visit by a care professional that received the appropriate training. Assessors were encouraged to use their own judgement and to use all sources of information available to complete the BelRAI instruments. Specifically, this means all assessors were told to rely on their own observations and to speak with the person being assessed and their family members and friends (if available) as well.

Approval to conduct this study was obtained from the Social and Societal Ethics Committee of KU Leuven (file number G-2019 05 1654). All potential participants were provided with full information about the study. In addition, all persons receiving or requesting social care services, or their representatives, were required to complete a written informed consent agreement before the start of the assessment. Refusal did not affect the care provided to the person.

### 2.5. Measures

#### 2.5.1. Dependent Variables

The six scores that were used as dependent variables in this study originated from the BelRAI Screener assessment. Based upon 41 items of the BelRAI Screener, standardized, reliable and validated scores can be calculated to determine a person’s functional status (measured by the interRAI Activities of Daily Living Hierarchy scale (ADLH) and Instrumental Activities of Daily Living Performance scale (IADLP) [51]), cognitive functioning (interRAI *Cognitive Performance Scale 2* (CPS2) [52]), and the presence of behavioural problems (six interRAI items) and psychological problems (five interRAI items) [46]. The total BelRAI Screener score ranges from 0 to 30, while the scores on each subscale range from 0 to 6, with higher scores denoting greater care needs or behavioural/psychological problems.

#### 2.5.2. Independent Variables

We included four indicators to investigate the association between assessor characteristics and care assessment scores. Regarding assessors’ socio-demographic characteristics, we included their *gender* and the *province* in which they were active at the time of the study. We added provinces to ensure that the variation found in care-assessment scores was not due to the geographical distribution of care professionals across Flanders. We also included two study-specific indicators:the number of assessments that were completed by the assessor andthe number of contact moments assessors had with trainers and other trainees.

As mentioned, assessors were trained prior to the data collection. They were also able to have a maximum of four contact moments with the trainers during the data collection.

The client-level indicators are part of the BelRAI Social Supplement assessment. It is important to emphasize that the responses on these items were not used to calculate the dependent variables (the BelRAI Screener scores).

A first set of BelRAI Social Supplement indicators included are about environmental characteristics of clients: the number of adults living with the client (client is included), the number of underage children living with the client, living situation (owner, renter, other), and the number of residential issues (sum of disrepair of the home, squalid condition, inadequate heating or cooling, lack of personal safety and limited access to home/rooms).

For civic engagement, we focus on the client’s communication skills with an indicator regarding their proficiency in Dutch (the native language of Flanders). To construct this indicator, we conduct a factor analysis with oblimin rotation on four items about the client’s ability to understand, speak, read, and write Dutch (0 = not or barely; 5 = (nearly) native language). The factor analysis reveals a single component with high internal reliability (Cronbach’s alpha = 0.89; for more information refer to Table A1).

A third type of indicators focus on psychosocial wellbeing: perceived financial situation (1 = make ends meet with great difficulty through to 6 = make ends meet very easily) and feelings of depression. For feelings of depression, we conducted a factor analysis with oblimin rotation on three self-constructed items. These assess whether respondents felt sad, anxious, and showed little interest in things they usually enjoy (0 = not in the last three days through to 3 = daily in the last three days). This too revealed a single component with high internal reliability (Cronbach’s alpha = 0.75; for more information refer to Table A2).

A final set of indicators provides information on informal care:whether the client received informal care (yes/no),whether they provide informal care to someone else (yes/no),whether an informal caregiver was present during the assessment (yes/no).

### 2.6. Analytic Strategy

The data were analysed using multilevel modelling (MLM) with maximum likelihood estimation using the PROC MIXED procedure in SAS Version 26 (SAS Institute, Raleigh, NC, USA) [53]. Only fixed effects are included. A common problem with MLM is that the N at the group level is often too low (around 20 to 30 in most studies), resulting in a high degree of parameter and standard error bias [54]. In order to “detect large structural effects at the between-level, at least 60 groups are required. To have an acceptable probability of detecting smaller effects, more than 100 groups are needed” [54] (p. 45). Given that our client-data were nested within 92 assessors, we were able to find moderate to small assessor-effects on BelRAI Screener scores.

Firstly, we calculated the intra-class correlation coefficient for each dependent variable to illustrate how much of the variance of the care scores can be attributed to the assessor-level [53]. In order to provide an overview of the association of client- and assessor-level variables with assessment scores, we conducted six different MLMs. In each model, we swapped out the dependent variables: overall BelRAI Screener score, IADLP, ADLH, CPS2, psychological problem scores, and behavioral problem scores. This allowed us to show which—if any—elements of the BelRAI Screener are associated with the BelRAI Social Supplement indicators. All variables were standardized by z-transformation.

In addition, we conducted a robustness check by removing the indicator regarding feelings of depression from the analysis, given its large number of missing values (n = 95) when compared to other study variables. This is likely because assessors were not obliged to ask these questions and may have opted not to do so in certain cases considering their sensitive nature. The results of this robustness check are not described, as the results yielded no differences with the main analyses, but they are shown in Table A3. Another robustness check was conducted to address concerns of a low number of clients per assessor, since this may result in biased standard errors of the lower level parameters and might also lead to failure to find group-level effects. In Table A4, we ran the same multilevel model as in our main analysis, but only included assessors with at least 10 clients (*N_i_* = 558; *N_j_* = 54), the minimum number of clients requested. The results of this robustness check closely align with those from the main analyses.

Finally, it is important to mention that we were unable to control for some basic demographic information of participants (e.g., gender, age, educational attainment) as these data were not collected due to ethical considerations.

## 3. Results

Our data were collected from December 2018 to December 2019, resulting in a sample of *N_i_* = 743 social care services clients nested in *N_j_* = 92 assessors. Eight assessors were unable to recruit clients to participate in the study. Each social care service client was assessed during a home visit using a BelRAI Screener and BelRAI Social Supplement. Although assessors were asked to complete a total of 10 assessments, most of them did not manage to reach this number, while a few exceeded it. A descriptive overview of the sample can be found in Table 1, and Pearson correlations between the dependent variables and key study variables in Table 2.

The results in Table 3 indicate that a sizable share of the variance of the overall BelRAI Screener score and several of the subscores can be attributed to clustering at the assessor-level, as illustrated by the intra-class correlation coefficients (ICCs). For each dependent variable, the ICC is calculated from a null model: only the dependent variable is added, while no independent variables are included at this point. The highest ICCs are found for the IADLP and ADLH-scores, for which 17% and 13% of the variance in these scores is due to variance between assessors, respectively. The overall BelRAI Screener score and the CPS2 scores are characterized by 11% and 10% ICCs, respectively. The scores for psychological (6%) and behavioural problems (5%) have the lowest ICCs.

When we look at the association of client-level and assessor-level variables with care assessment scores using multilevel modelling (Table 4), a variety of patterns emerged. When looking at client-level variables, we consider indicators that corresponded to the four themes of the BelRAI Social Supplement:environmental assessment,civic engagement,psychosocial wellbeing, andinformal care and support [48].

As for environmental characteristics, we observe that the number of adults living with the client is positively associated with care scores for the BelRAI Screener overall (*b* = 0.10, *p* < 0.05), IADLP (*b* = 0.16, *p* < 0.001), and behavioural problem scores (*b* = 0.07, *p* < 0.05). However, the number of children living with the client is negatively associated with scores for the BelRAI Screener, IADLP, ADLH, and the CPS2. It is unclear whether this latter effect reflects lower care needs for these individuals or if this is a hidden age effect in the sample, given that we are unable to control for this factor. While there are no clear effects of an individual’s living situation on care scores, individuals who owned their residence and are still paying off their loan tend to receive lower care scores than owners without loans. Having residential issues is associated with greater psychological (*b* = 0.10, *p* < 0.01) and behavioral problems (*b* = 0.15, *p* < 0.001).

Looking at the role of Dutch proficiency, which is used here as a proxy for civic engagement, the relationship is clear: individuals who are proficient in Dutch receive consistently lower care scores than those who are less proficient, except for the ADLH—which is to be expected, as this scale considers strictly physical activities that an individual can perform. The perceived financial situation, which reflects psychosocial wellbeing, is associated with lower psychological problem scores only (*b* = −0.10, *p* < 0.05). Feelings of depression are associated with higher CPS2 (*b* = 0.10, *p* < 0.01), psychological (*b* = 0.38, *p* < 0.001), and behavioural problem scores (*b* = 0.10, *p* < 0.01), along with a higher overall BelRAI Screener score (*b* = 0.22, *p* < 0.001).

Finally, looking at informal care indicators, we find that individuals who receive informal care have higher IADLP (*b* = 0.19, *p* < 0.05) and ADLH (*b* = 0.21, *p* < 0.05) scores. Those who provide informal care receive lower IADLP (*b* = −0.31, *p* < 0.01) and ADLH (*b* = −24, *p* < 0.05) scores, and a lower overall BelRAI Screener score (*b* = −0.19, *p* < 0.05). Finally, having an informal caregiver present during the assessment is associated with higher BelRAI Screener (*b* = 0.31, *p* < 0.001), IADLP (*b* = 0.37, *p* < 0.001), and CPS2 (*b* = 0.33, *p* < 0.001) scores.

At the assessor-level, we find that the number of assessments conducted is associated with higher BelRAI Screener (*b* = 0.11, *p* < 0.05), IADLP (*b* = 0.11, *p* < 0.05), and CPS2 (*b* = 0.08, *p* < 0.05) scores. The other assessment-level indicators yield few significant results, from which no clear pattern emerged.

## 4. Discussion

Care providers often use care assessments and their output to gather information on the effectiveness of treatments. It also facilitates comparisons with other clients who have complex care needs [11]. It is important to analyse the quality of the data, as this affects all users of the assessments [12,13]. Examination of whether and how care assessment scores vary between assessors is one important element to know if assessor-related bias is present in the data. Additionally, a closer look at which socio-psychological factors are associated with certain assessment scores can yield information about the socio-economic and socio-psychological context in which the person is living. In this study, we investigated to what extent some socio-economic and psychological determinants on the client-level and assessor-related determinants were associated with BelRAI Screener care assessment scores. We used data with 743 community-dwelling adults in Flanders, Belgium nested within 92 assessors.

Even though the assessors—mainly women, in line with numbers in other countries —received standardized training to help them learn the clear coding guidelines in order to uniformly assess and score responses, our data show signs of variance clustering on the assessor-level. By looking at the intra-class correlation coefficients (ICCs), we can conclude that an ample share of the variance of the overall BelRAI Screener scores (11%) and several of the BelRAI Screener subscores of community-dwelling adults in Flanders was due to variance between assessors. This variance is not entirely unexpected, as both BelRAI Screener and BelRAI Social Supplement assessments were new to the assessors and were completed during a single home visit. Research shows that face-to-face assessments generally produce larger interviewer variance as the assessor’s own beliefs and attitudes can influence what is observed. On the participants’ side, the presence of an assessor in the home setting can influence their behavior and answers [43,44,45,46,47,48,49,50,51,52,53,54,55].

Our results show a significant positive association between the presence of a third party during the assessment and the overall BelRAI Screener score and various subscores. Aquilino [56] proposes that third party effects may depend on factors related to both the question content and the person(s) involved. For example, findings may be different on questions concerning factual information than those on attitudes or sensitive information. This study only looked at variance in the (sub)scores of the BelRAI Screener instrument. While the items in the BelRAI Screener focus on physical functioning and thus factual information, the BelRAI Social Supplement also includes some items on the person’s feelings and other sensitive topics such as their subjective financial situation. Further research should look at the interaction between the presence of informal caregiver and BelRAI Social Supplement items dealing with sensitive topics (such as symptoms of depression and satisfaction about the informal caregiving situation), as other research indicates that the presence of a third party during a survey “does not compromise data quality but may in fact improve it” [57,58] (p. 18).

During the training cycles, assessors were asked to participate in several discussion groups. The assessors participating in the discussion groups had mixed opinions on the presence of an informal caregiver or third party during the assessments. Some assessors preferred the presence of a third party to keep the conversation flowing, while others indicated that the presence of a ‘well-informed’ family caregiver can steer the conversation too much, in which case the person with care needs is no longer being heard adequately. These differing opinions on third party presence is also shown when the assessor is talking with the clients who do not want to admit their weaknesses and overestimates themselves, or, to the contrary, clients who may feel very lethargic and underestimate themselves. A close informal caregiver may be able to help clarify the narrative for the assessor. When a third party is far removed from the actual care situation or has ulterior motives such as solely obtaining cash benefits, the possibility of over- or underestimation during an assessment is real. These findings from practice are supported by the notion that third-party effects can depend on the type of third person involved [56,58].

Our analyses identify correlations between the BelRAI Screener (sub)scores and some of the social context indicators assessed in the BelRAI Social Supplement. This implies that the impact of socio-psychological and environmental factors on a person’s physical functioning is partially contained in the BelRAI Screener (sub)scores. Corresponding with previous research, we find that the client’s level of Dutch proficiency is negatively associated with care scores, while the number of residential issues (housing conditions) is positively associated with care scores, [36,59,60]. Concretely, this means that a higher number of residential issues corresponds with a higher dependency score, thus more negative health outcomes. However, a large number of our client-level and assessor-level indicators do not explain the variance found in the BelRAI Screener (sub)score.

To summarize, our findings show that there is significant variation between assessors in terms of care scores. Although some variation is to be expected, given that some regions may contain a greater number of persons with high care needs than other regions, for example, there is still some cause for concern. Although the development of new and innovative care instruments is of paramount importance to assess (a change in) care needs, these instruments may lose much of their value if assessors cannot score care needs in a (sufficiently) standardized way. If assessors vary widely in their scoring methods, then individuals who deserve certain benefits may be left out, or some may receive benefits when they should not. Few assessor-related factors were associated with care scores in the current study, but we recommend that the training of assessors be expanded and monitored closely, for example through regular discussion groups.

We also found that various socio-psychological and contextual factors are linked with care scores. This highlights the growing need to consider care needs within this broader, holistic context, and move beyond this strict biomedical model of disability. In addition to the scores from assessments focused on primarily biomedical issues like the BelRAI Screener, data from instruments like the BelRAI Social Supplement may also be used to in the decision to allocate certain welfare benefits in the future. Although this is currently not a common practice, the growing adoption of the WHO’s framework on integrated people-centered health services, in which a person with disabilities is surrounded by their family and close community, may enable such policy-relevant links to be made. Based on this study, the strong link between certain socio-psychological factors and care needs certainly indicates that this could be an important way in which care services can provide an increasingly fair system for all. Three limitations of our study should be noted. First, our sample of assessors or clients is not selected at random due to time and privacy constraints. The incomplete socio-demographic information of our sample makes it impossible for us to confirm the representativity for all home care clients. As mentioned earlier, we were also unable to control for these basic characteristics in our analyses. Nevertheless, through the BelRAI Social Supplement we collected information on other social context characteristics. These show that our sample is diverse, with an exception of families with children present in the household and persons with a low level of Dutch proficiency. Secondly, the data collected on the assessor-level is limited. Both role-restricted and role-independent interviewer effects are hard to define within our data. Gaining insight into the conduct of the assessor towards the client during the selection of possible respondents and the actual assessment is difficult [45,61]. However, gathering information on role-independent interviewer aspects is more accessible. In this study, we have no information on the level of experience in the job as professional assessor, nor their attitudes (e.g., welfare deservingness). Those two interviewer characteristics were specifically cited as potentially having an influence on the outcomes in a survey [43,45,55,62,63]. It is reasonable to believe that the assessor’s welfare deservingness attitudes will influence their scoring on the BelRAI Screener instrument as it is used concurrently for care planning and allocating regional care budgets [46,64]. Following this logic, an assessor who reports high conditionality may be quicker to focus on the person’s ability to be independent, while an assessor who reported low conditionality will focus on the aspects where care and support is needed [65]. Thirdly, it could be argued that the presence of an informal caregiver during the assessment (which is the case for a little over 50% of the current sample) may contribute to biased scores to some extent. In some cases (e.g., a lack of language proficiency or the presence of cognitive problems), this presence may be unavoidable, but it may be less (if at all) necessary in other cases. Future studies should keep such considerations in mind when collecting data on care needs. Since June 2021, the BelRAI Screener instrument is being used in all social care services in Flanders to get a first assessment of a person’s care needs. An optimized version of the BelRAI Social Supplement will be implemented in Flanders starting from June 2022 [66]. This region-wide implementation of the BelRAI instruments will result in additional longitudinal, representative and comprehensive data at the client and assessor-level of the broad social care services clientele. These data will facilitate further research into the intra-rater reliability, as well as test-retest reliability [12,65]. Supplementing this rich data with information on the assessor’s characteristics is necessary to have a better understanding of the fairness of the current policies regarding the use of BelRAI instruments in Flemish home care, namely to examine if the use of standardized assessments will indeed create a system that “responds equally well to everyone, without discrimination or differences in how people are treated” [9] (p. 26). Further studies using multiple assessors with one client can in turn facilitate further research into inter-rater reliability [12,66,67,68].

## 5. Conclusions

This paper describes the results of our investigation into what extent care assessment scores given by social care services to their community-dwelling clients in Flanders, Belgium are associated with client-related and assessor-related factors. We use a rich dataset which includes information on both a person’s care needs via the BelRAI Screener instrument and their social context via the BelRAI Social Supplement assessment. Data on these detailed client-level indicators are supplemented by a handful of assessor-level indicators. Our sample consists of 743 community-dwelling adults and 92 assessor from the social care services in Flanders, Belgium. Social care at home is a highly accessible service in Flanders and serves a broad clientele. Our sample consists of community-dwelling adults with chronic care needs, and excludes persons solely receiving maternity care and/or services for families in precarious situations [48]. Our findings show that variance in care assessment scores can be attributed to certain assessor-level and other client-level indicators. This is important to consider when using this data for the fair allocation of health services and/or care budgets.

## Figures and Tables

**Table 1 ijerph-18-11845-t001:** Descriptive overview of client (*n* = 743) and assessor (*n* = 92) indicators.

	Frequency		Percentage	
**Clients** (*n* = 743)				
** *Living situation* **				
Owner without loan	343		46.2	
Owner with loan	54		7.2	
Renter with private owners	149		20.1	
Renter social residence	142		19.1	
Other	53		7.1	
*Missing*	*2*		*0.3*	
** *Provide informal care* **	127		17.1	
*Missing*	*10*		*1.3*	
** *Receive informal care* **	585		78.7	
*Missing*	*3*		*0.4*	
** *Informal caregiver present* **	374		50.3	
*Missing*	*3*		*0.4*	
**Assessors** (*n* = 92)				
** *Gender* **				
Male	10		10.9	
Female	82		89.1	
** *Province* **				
Antwerp	19		20.7	
East Flanders	23		25.0	
Flemish Brabant	4		4.3	
Limburg	19		20.7	
West Flanders	21		22.8	
Other	6		6.5	
	Min	Max	Mean	SD
**Clients** (*n* = 743)				
** *# residing adults* **	1	7	1.57	0.73
** *# residing children* **	0	5	0.14	0.53
** *Residential issues* **	0	5	0.50	0.88
*Missing*	*n = 9*			
** *Dutch proficiency* **	0	5	4.23	1.21
*Missing*	*n = 12*			
** *Perceived financial situation* **	1	6	3.61	1.20
*Missing*	*n = 10*			
** *Feelings of depression* **	0	3	1.13	1.00
*Missing*	*n = 95*			
**Assessors** (*n* = 92)				
** *Number of assessments* **	1	20	8.21	3.39
** *Number of contact moments* **	1	4	3.13	0.93

**Table 2 ijerph-18-11845-t002:** Mean scores and Pearson correlations between dependent variables and key study variables.

	M (SD)	1.	2.	3.	4.	5.	6.	7.	8.	9.	10.	11.	12.
1. BRS	7.90 (4.30)	1											
2. IADLP	3.41 (1.31)	0.69 **	1										
3. ADLH	1.81 (1.48)	0.56 **	0.56 **	1									
4. CPS2	1.32 (1.56)	0.75 **	0.43 **	0.14 **	1								
5. Psychological problems	0.89 (1.36)	0.52 **	−0.00	−00.05	0.33 **	1							
6. Behavioral problems	0.47 (1.17)	0.60 **	0.15 **	0.03	0.40 **	0.40 **	1						
7. Residential issues	0.50 (0.88)	0.11 **	−0.06	0.02	0.02	0.22 **	0.16 **	1					
8. Dutch proficiency	4.23 (1.21)	−0.35 **	−0.26 **	−0.04	−0.38 **	−0.15 **	−0.26 **	−0.05	1				
9. Perceived financial situation	3.61 (1.20)	−0.02	0.11 **	0.02	0.03	−0.23 **	0.01	−0.21 **	0.04	1			
10. Feelings of depression	1.13 (1.00)	0.28 **	0.07	0.07	0.13 **	0.42 **	0.16 **	0.16 **	−0.05	−0.31 **	1		
11. Number of assessments	9.45 (2.52)	0.13 **	0.11 **	0.08 *	0.04	0.11 **	0.06	0.04	−0.01	−0.07	0.10 *	1	
12. Number of contact moments	3.20 (0.87)	−0.02	0.05	0.04	−0.05	−0.06	−0.04	0.02	0.01	−0.03	−0.06	0.13 **	1

Note: ** *p* < 0.01; * *p* < 0.05. BRS = BelRAI Screener Score; IADLP = Instrumental Activities of Daily Living Performance scale; ADLH = Activities of Daily Living Hierarchy scale; CPS2 = Cognitive Performance Scale 2.

**Table 3 ijerph-18-11845-t003:** Intra-class correlation coefficients of dependent variables in null models.

	ICC	AIC	−2LL
BelRAI Screener score	0.11	2122.3	2118.3
IADLP	0.17	2096.6	2092.6
ADLH	0.13	2110.3	2106.3
CPS2	0.10	2119.2	2115.2
Psychological problems	0.06	2135.3	2131.3
Behavioral problems	0.05	2138.2	2134.2

Note. ICC = intra-class correlation coefficient; AIC = Akaike Information Criterion; −2LL = −2 Log Likelihood.

**Table 4 ijerph-18-11845-t004:** Multilevel regression analyses of client- and assessor-level variables on assessment scores.

	BRS	IADLP	ADLH	CPS2	Psychological	Behaviour
	B (se)	B (se)	B (se)	B (se)	B (se)	B (se)
**Intercept**	−0.21 (0.17)	−0.09 (0.19)	0.13 (0.22)	−0.48 * (0.19)	−0.04 (0.18)	−0.17 (0.15)
**Individual indicators**						
** *Environmental characteristics* **						
Number of adults living with client	0.10 * (0.04)	0.16 *** (0.04)	0.06 (0.04)	0.04 (0.04)	−0.01 (0.04)	0.07 * (0.04)
Number of children living with client	−0.15 *** (0.03)	−0.14 *** (0.04)	−0.13 ** (0.04)	−0.09 * (0.04)	−0.06 (0.04)	−0.04 (0.03)
** *Living situation (ref: owner without loan)* **						
Owner with loan	−0.23 * (0.12)	−0.33 * (0.14)	−0.34 * (0.15)	−0.24 (0.13)	0.27 (0.14)	−0.05 (0.12)
Renter with private owners	0.04 (0.09)	−0.02 (0.10)	0.03 (0.11)	0.14 (0.10)	0.04 (0.10)	−0.12 (0.09)
Renter social residence	0.02 (0.09)	0.03 (0.10)	−0.17 (0.11)	0.02 (0.10)	0.23 * (0.10)	−0.08 (0.09)
Other	0.23 (0.15)	0.11 (0.17)	0.19 (0.18)	0.39 * (0.16)	0.11 (0.17)	−0.14 (0.15)
Residential issues	0.12 ** (0.03)	0.04 (0.04)	0.08 * (0.04)	−0.00 (0.04)	0.10** (0.04)	0.15 *** (0.03)
** *Civic* ** ** *engagement* **						
Dutch proficiency	−0.26 *** (0.04)	−0.19 *** (0.04)	−0.02 (0.04)	−0.34 *** (0.04)	−0.12 ** (0.04)	−0.10 ** (0.04)
** *Psychosocial wellbeing* **						
Perceived financial situation	−0.03 (0.04)	0.06 (0.04)	−0.04 (0.05)	0.01 (0.04)	−0.10 * (0.04)	−0.01 (0.04)
Feelings of depression	0.22 *** (0.03)	0.05 (0.04)	0.04 (0.04)	0.10 ** (0.04)	0.38 *** (0.04)	0.10 ** (0.03)
** *Informal care indicators* **						
Provide informal care	−0.19 * (0.08)	−0.31 ** (0.09)	−0.24 * (0.10)	−0.06 (0.09)	0.04 (0.09)	−0.01 (0.08)
Receive informal care	0.14 (0.09)	0.19 * (0.10)	0.21 * (0.10)	0.10 (0.09)	−0.04 (0.09)	−0.08 (0.08)
Informal caregiver present	0.31 *** (0.07)	0.37 *** (0.08)	0.15 (0.09)	0.33 *** (0.08)	0.01 (0.08)	0.07 (0.07)
**Assessor** **indicators**						
** *Gender (ref: female)* **						
Male	−0.02 (0.13)	−0.15 (0.15)	−0.13 (0.17)	0.17 (0.15)	−0.04 (0.14)	0.08 (0.11)
Number of assessments	0.11 * (0.04)	0.11 * (0.05)	0.08 (0.06)	0.04 (0.05)	0.08 * (0.05)	0.04 (0.04)
Number of contact moments	0.02 (0.04)	0.07 (0.05)	0.05 (0.05)	−0.01 (0.05)	−0.03 (0.04)	−0.05 (0.04)
** *Province (ref: East Flanders)* **						
Antwerp	−0.21 (0.12)	−0.17 (0.13)	−0.45** (0.15)	−0.02 (0.13)	−0.04 (0.12)	0.06 (0.10)
Flemish Brabant	−0.21 (0.22)	−0.60* (0.25)	−0.32 (0.28)	0.08 (0.25)	0.04 (0.23)	0.16 (0.19)
Limburg	−0.12 (0.12)	−0.17 (0.14)	−0.18 (0.16)	−0.05 (0.14)	−0.03 (0.13)	0.13 (0.11)
West Flanders	0.12 (0.12)	0.05 (0.14)	−0.19 (0.16)	0.19 (0.14)	0.15 (0.13)	0.20 (0.10)
Other	−0.56 ** (0.20)	−0.05 (0.23)	−0.38 (0.26)	−0.42 (0.23)	−0.61 ** (0.22)	−0.28 (0.18)
**Variance** **components**						
*Level 2: Assessor*	0.05 **	0.07 **	0.11 **	0.08 **	0.05 *	0.02
*Level 1: Individual*	0.55 ***	0.68 ***	0.80 ***	0.62 ***	0.70 ***	0.56 ***
**AIC**	1476.4	1610.3	1711.9	1556.6	1611.7	1470.5
**−2 Log Likelihood**	1472.4	1606.3	1707.9	1552.6	1607.7	1466.5

Note: *** *p* < 0.001; ** *p* < 0.01; * *p* < 0.05. *N_i_* = 614; *N_j_* = 92. BRS = BelRAI Screener Score; IADLP = Instrumental Activities of Daily Living Performance scale; ADLH = Activities of Daily Living Hierarchy scale; CPS2 = Cognitive Performance Scale 2.

## Data Availability

The data that support the findings of this study are available from the Flemish Policy Research Center for Welfare, Public Health and the Family (Flemish Acronym: SWVG) but restrictions apply to the availability of these data, which were used under license for the current study, and so are not publicly available. Data are however available from the authors upon reasonable request and with permission of the Flemish government.

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
