# Peer review of "Examining Care Assessment Scores of Community-Dwelling Adults in Flanders, Belgium: The Role of Socio-Psychological and Assessor-Related Factors"

_ijerph, 2021, doi:10.3390/ijerph182211845_

Round 1

Reviewer 1 Report

It is complex to establish comparisons between different countries on how to evaluate social and health services. The initiative of this study can help to develop more adequate valuation methods that allow standardizing the way of financing services by public institutions.

In particular when there are behavioral problems, testimonials from caregivers will be essential.
I consider it necessary to include among the limitations of the research the need to differentiate whether people with cognitive problems should be evaluated in the presence of other caregivers and, above all, which people should be interviewed to know the quality of the services that people receive dependents and what needs are most urgent.

Reviewer 2 Report

The data analysis portion is clear and solid. However, I want the authors to revise Introduction and Discussion.

Introduction - This article will be much stronger with discussion of the background. I was left with some fundamental questions, "Why does this topic matter in Belgium?" "What are the concerns/problems people in Belgium have under the current system?" This discussion would help readers outside of Belgium/Europe understand the meaning of this paper better.

Discussion - What does these findings mean? What kind of reform/changes would the authors recommend? Would they be possible? (Clearer discussion on the implication of the findings)

------------------

I would like the authors to contextualize the data and findings with more detailed discussion of social, political and economic environment of contemporary Belgium.  

Reviewer 3 Report

Referee report for “Examining care assessment scores of community-dwelling adults in Flanders, Belgium: The role of socio-psychological and assessor-related factors”; Manuscript ID: ijerph- 1419179 for IJERPH

This study examines a potential bias caused by health raters on social care clients’ health scores, as well as other socio-psychological determinants that might affect these ratings. Although I feel that these could be two different studies the authors combine them into one analysis (model). Either way, this is an interesting paper that address an important topic with serious health policy implications. My suggestions/concerns are the following:

Lines 60-63. I would add also, to better train assessors and form health policies. 

Line 71. Please explain why you think it is impossible to compare the outcomes, if the questionnaire is common and validated, it seems entirely possible.

Line 77. Authors reference a page but cite 4 papers. To which it corresponds?

Line 156. I believe it would benefit the paper if somewhere in the text, the relationship between clients and social care services was explained in a few words as a background context.

Line 162. Why was this population excluded?

Lines 171-172. Acronyms not previously defined in text.

Line 192-195. It is not clear to me what educational/occupational background “care professionals” have. Where there any criteria for their recruitment? Were they tested after their training in any way to make sure they can assess health properly? Also, I would like some information about the nature of the organisations that employ them. Are they for profit? Do they have clear financial incentives to “under-diagnose” their clients?

Line 200. Is there a chance that clients are from the same household? Because this dependence violates assumptions and needs to be modelled ideally.

Line 243. Were those self-reported or verified?

Line 265. The last 3 days may not be representative of their usual feelings.

Lines 255-259. Since the authors are interested in identifying a latent variable, that is, Dutch proficiency, I do not understand why PCA was chosen over factor analysis, or is it a combination?  I understand the use of oblimin rotation because of the correlation between variables, but wasn’t the purpose of the PCA to extract a single component the maximises the variance so it can be used on the model as an explanatory variable instead?

Line 273. Please provide more details on the model specification.

Lines 293-296. This may not be the authors’ fault, but since when absolute essential individual-level information, collected in pretty much every survey, pose ethical concerns? The information collected for the assessment contains much more sensitive information than gender and age. Failing to control for age especially, is a big problem when you try to model health outcomes.

Lines 299-300. From what I understand there was no allocation of assessors to clients. What I do not understand is the laissez-faire policy of the organisations regarding the matter. Why spent money training assessors and do nothing to ensure that these resources are not wasted. I feel the authors need to elaborate more and criticise how home care assessment works in Flanders.

Lines 302-303. See, this causes problems in multilevel models. If you don’t have enough observations on the lower level (i.e. clients), this will lead to biased standard errors of the lower level parameters and might also lead to failure to find group-level effects. Since the number of groups is fairly large, maybe the use of robust (Huber-White) standard errors might save the day. Again, the fact that authors do not elaborate on the MLM does not allow me to help more. Fixed and random effects behave differently and I do not have information on the standard errors used either. I provide some bibliography at the end to assist, should authors decide to stick with the MLM.

Line 306 Table 1. Females appear to dominate. Does literature suggest a gender bias in rating peoples health? How these affect the internal and external validity of the study? In other words, is this pattern representative of assessors in Flanders and assessors in other areas/countries? Please include some comments in the Discussion section.

Lines 313-320. More information is needed for the interpretation of the ICC analysis (i.e. which model was used and why, consistency or absolute agreement, individual (client) or assessor average coefficients). The fact that each client was assessed from only one assessor really limits this methodology for identifying assessor bias, especially since neither was random. Assessor average correlations can be heavily confounded by all sort of factors.

Line 321. typo AIC

Lines 516-527. Please add to the notes of these tables the level of significance denoted from the stars. 

General comment

The abstract of the paper can be improved.

Bibliography

Maas, C.J.M. and Hox, J.J. (2004), Robustness issues in multilevel regression analysis. Statistica Neerlandica, 58: 127-137. https://doi.org/10.1046/j.0039-0402.2003.00252.x

Moineddin, R., Matheson, F.I. & Glazier, R.H. A simulation study of sample size for multilevel logistic regression models. BMC Med Res Methodol 7, 34 (2007). https://doi.org/10.1186/1471-2288-7-34

Theall KP, Scribner R, Broyles S, et al. Impact of small group size on neighbourhood influences in multilevel models. Journal of Epidemiology & Community Health 2011;65:688-695.

Round 2

Reviewer 2 Report

I appreciate all the edits. The discussion became much stronger, and the paper flows much smoother. The background could be expanded for the audience who are not familiar with the setting (i.e., audience outside of Eruope).  

Reviewer 3 Report

I appreciate the hard work of the authors to address all my comments. I am glad I could help improve this important study.